# Executive function and the continued influence of misinformation: A latent-variable analysis

Paul McIlhiney *, Gilles E. Gignac, Ullrich K. H. Ecker, Briana L. Kennedy, Michael Weinborn

School of Psychological Science, University of Western Australia, Crawley, WA, Australia

* paul.mcilhiney@research.uwa.edu.au

## Abstract

Misinformation can continue to influence reasoning after correction; this is known as the continued influence effect (CIE). Theoretical accounts of the CIE suggest failure of two cognitive processes to be causal, namely memory updating and suppression of misinformation reliance. Both processes can also be conceptualised as subcomponents of contemporary executive function (EF) models; specifically, working-memory updating and prepotent-response inhibition. EF may thus predict susceptibility to the CIE. The current study investigated whether individual differences in EF could predict individual differences in CIE susceptibility. Participants completed several measures of EF subcomponents, including those of updating and inhibition, as well as set shifting, and a standard CIE task. The relationship between EF and CIE was then assessed using a correlation analysis of the EF and CIE measures, as well as structural equation modelling of the EF-subcomponent latent variable and CIE latent variable. Results showed that EF can predict susceptibility to the CIE, especially the factor of working-memory updating. These results further our understanding of the CIE's cognitive antecedents and provide potential directions for real-world CIE intervention.

## Introduction

During the Australian "Black Summer" wildfires in 2019/2020, some groups repeatedly claimed that arson caused the fires; while these claims were later debunked, some continued to falsely believe in arson's causative role, contributing to further polarisation in the climate-change debate [1,2]. Such a phenomenon is a real-world example of the *continued influence effect* (CIE) [3,4]. Specifically, the CIE constitutes the continued use of information in inferential reasoning after said information has been retracted or corrected. Psychological lab studies have demonstrated that the CIE can occur even when materials are fictional, suggesting that cognitive factors play a significant role in the CIE [e.g., 4–8]. Consequently, there has been two decades worth of research attention on the role that cognitive processes play in the CIE [e.g., 7,9–15; also see 16]. However, little is known about how individual differences in cognition influence CIE susceptibility. While recent individual differences research has suggested a potential role for working-memory updating [17], attempts to replicate these results have

**Data Availability Statement:** All relevant data and supporting files are available from the OSF database (https://osf.io/hw47f/).

**Funding:** PM's contribution to this research was supported by an Australian Government Research Training Program (RTP) Scholarship. UKHE was supported by grant FT190100708 from the Australian Research Council. The funders had no role in study design, data collection and analysis, decision to publish, or preparation of the manuscript.

**Competing interests:** The authors have declared that no competing interests exist.

failed [18] and so require further investigation. One model that may be useful in this regard is Miyake's model of executive function [19], which includes working-memory updating as a subcomponent alongside prepotent-response inhibition and mental-set shifting. Using Miyake's model, understanding of the relationship between cognitive abilities and the CIE may be improved; that is, results could provide further insight into current cognitive theories of the CIE and aid intervention efforts in the real world.

One of the two theories proposed to explain the role of cognitive processes in the CIE is the mental-model-updating account. This account is derived from mental-model theory, which postulates that people build mental models of events in real time, and update these models when new information (e.g., a correction) is received [20]. From this perspective, a CIE may arise from a failure in memory-updating processes [16,21]. Supporting this theory, Ecker et al. [11] demonstrated that retractions reduced the CIE more effectively when original misinformation was repeated (versus not being repeated) at the time of retraction. The authors suggested that repeating misinformation may increase the salience of retractions, thus aiding their integration into the mental model. In a further study, neural pathways associated with memory updating showed greater activation during processing of corrections than during processing of non-corrective information [22; however, see 23]. This account suggests that better updating of working memory may promote better integration of corrections into mental event models. In line with this thinking, working-memory capacity has been shown to be predictive of the CIE [17].

An alternative, complementary theory is a retrieval-based account, which postulates that both misinformation and correction are stored concurrently and compete for activation at memory retrieval [e.g., 12,24]. This account suggests that continued influence can occur if misinformation is selectively retrieved by an automatic familiarity-driven process but fails to be inhibited when responding to event-related test questions [12]. In support of this account, Swire et al. [25] demonstrated that factors that are theoretically conducive to use of familiarity-based retrieval (as opposed to more strategic, recollection-based retrieval, e.g., longer study-test delays; advanced participant age) were associated with greater reliance on corrected misinformation. This account suggests that greater capacity for inhibition of prepotent responses may translate to an enhanced ability to inhibit responses based on automatically retrieved misinformation. While no previous research has provided empirical evidence for a link between inhibitory processes and the CIE, work in the knowledge-revision literature has shown that better prepotent-response inhibition provides a mechanism to manage interference from misconceptions when reading accurate but counterintuitive statements [26].

More recent efforts have been made from an individual-differences perspective to understand the role of cognitive processes in the CIE; however, these efforts have been thus far limited to investigations of the CIE's relationship with working-memory capacity [17] and verbal cognitive ability [27]. Brydges et al. [17] investigated the relationship between CIE susceptibility and working-memory capacity. Participants were given several measures of working-memory capacity, from which a latent variable was derived and used to try predict performance on a CIE paradigm task [see 14]. Briefly, the CIE paradigm task involves presenting several news reports containing critical information that is, or is not, subsequently corrected; inferential-reasoning questions are then used to gauge participants' reliance on the critical (mis-)information. In line with Brydges et al.'s [17] predictions, higher working-memory capacity predicted lower CIE susceptibility [however, see 18]. The authors suggested that their results support the mental-model-updating account of the CIE, as model updating relies on working memory. De keersmaecker and Roets [27] investigated whether verbal cognitive ability predicted CIE susceptibility. The authors partly based their investigation on the comprehensive assessment of rational thinking model [28], which suggests that cognitive ability is relevant to the inhibition

and overriding of previously learned responses. Results showed that higher verbal cognitive ability predicted lower CIE susceptibility.

In sum, then, recent evidence suggests that individual differences in higher-level cognitive abilities—such as working-memory capacity [17]—may play a role in determining individual susceptibility to the CIE. However, this line of investigation has yet to assess both theoretically implicated cognitive processes—memory updating and inhibition—directly and concurrently. Thus, an exploratory investigation of the correlations between updating, inhibition, and CIE measures is warranted. Furthermore, both candidate cognitive processes can also be conceptualised as subcomponents of *executive function* (EF), as in Miyake's model [19; for reviews, see 29–31]. Therefore, assessing the relationship between the subcomponents of Miyake's EF model and CIE susceptibility may allow for a more nuanced understanding of how individual differences in cognitive abilities affect the CIE, as well as how well each CIE theory explains the CIE.

## The current study

The current, exploratory, study aimed to investigate whether individual differences in executive function (EF) were predictive of individual differences in CIE susceptibility. To do this, participants were given three measures of each EF subcomponent from Miyake et al.'s [19] model (i.e., working-memory updating, prepotent-response inhibition, and mental-set shifting), followed by a CIE-paradigm task [e.g., 32]. While we did not expect an effect of shifting, we included it for completeness. We used confirmatory factor analyses (CFA) to fit our data, starting with the model architectures suggested by Miyake and colleagues but also testing alternative models. Subsequently, we ran a correlation analysis between the EF tasks and the CIE task, as well as a structural equation model (SEM) analysis of the CIE-task latent variable regressed onto our EF model. Note that both verbal and non-verbal measures of each EF subcomponent were employed, as Brydges et al. [17] found that only verbal working-memory capacity measures correlated with CIE susceptibility; specifically, of the three measures we used for each EF subcomponent, two were verbal and one was non-verbal.

As the current study is exploratory, due to the limited individual-differences evidence for CIE theoretical accounts [17,27], we simply hypothesised that greater EF ability would predict lower CIE susceptibility. As such, we predicted that there would be: (i) negative observed-score correlations between one or more of the EF measures and CIE measure, and (ii) a significant negative β weight between one or more of the EF-subcomponent latent variables and the CIE latent variable. However, note that finding such relationships for updating and inhibition measures would provide some evidence for the mental-model-updating and selective-retrieval accounts of the CIE, respectively.

## Method

This study used a cross-sectional design with one independent variable (executive function; EF) that had three sub-dimensions (updating, inhibition, shifting) and one dependent variable (CIE susceptibility). Each sub-dimension of the independent variable was measured with three standardised tests to allow formation of a latent variable, which was used to predict CIE susceptibility, as measured by a standard CIE paradigm task. Our research was approved by the University of Western Australia's Human Research Ethics Office. Participants provided written informed consent after reading an information sheet.

### Participants

Participants were undergraduate students from the University of Western Australia (UWA), who participated for course credit. As we anticipated exclusions, and a minimum of 200

participants is recommended for SEM [33], we recruited 300 participants in total. Participants were excluded if performance on EF and CIE measures was suggestive of poor effort or engagement with the measures (e.g., below chance performance; see Materials section for details).

In total, 45 participants were excluded, with 34 exclusions from the EF tasks and eight exclusions from the CIE task due to poor performance; the remaining three exclusions resulted from the Flanker task being accidentally skipped, inattentiveness of one participant as observed by the experimenter, and a mock fire alarm. Thus, the final sample size was $N = 255$, with 55 men, 198 women, and two participants of undisclosed gender (mean age $M = 20.56$, $SD = 6.22$; age range: 18–53).

## Materials

**Updating tasks.** *Letter N-Back Task (Verbal).* To measure verbal updating ability, the *N*-Back task was employed [34], using 20 consonants as stimuli, based on Ragland et al. [35]. As with all EF tasks used in this study, we used the Inquisit 6 version of the Millisecond test library [36]. In each block, participants were presented with a sequence of 15 (white) letters that were shown one-at-a-time on a series of black screens. Each letter appeared in the middle of the screen for 500 ms, with a 2000 ms delay between letters. Participants were instructed to press the 'A' key on the keyboard when the current letter matched the letter three screens previous (i.e., a 3-back task), and to do nothing if the letters did not match. Responses were to be given as quickly and accurately as possible. One practice block was given first, wherein nine letters were presented including three targets (i.e., letters matching those three positions back). Subsequently, six main blocks were run with five target letters in each—a total of 30 target letters. Performance was determined by the overall proportion of correct responses. Testing time was approximately seven minutes. One participant was excluded based on their performance in this task, as they had a mean response time equal to 2500 ms, indicating that they did not respond at all.

*Keep-Track Task (Verbal).* The second task used to measure verbal updating ability was the keep-track task [37]. The version of the task we used was based on Friedman et al. [38] On each trial, participants were given a sequence of 15 words each belonging to one of six categories (i.e., animals, colours, countries, distances, metals, and relatives). Each word was presented in the middle of the screen for 2500 ms, with a 500 ms delay between words. Participants were instructed to remember the last word given from each included category, then report these words in a questionnaire at the end of each trial. A minimum of two words and a maximum of three words were presented from each category per trial. Nine trials were run wherein four category lists had to be updated (36 words in total). The words selected from each category were randomised, with repetitions not allowed. Updating ability was determined by the overall proportion of correct responses. Testing time was approximately seven minutes.

*Shape N-Back Task (Non-verbal).* The Shape *N*-Back task was essentially identical to the Letter *N*-Back task discussed above; however, there were some differences (based on Jaeggi et al. [39]). Firstly, the stimuli used in this task were eight irregular yellow shapes; each stimulus was presented for 500 ms with a 2500 ms delay between shapes. Secondly, since the shape task was inherently more difficult than the letter task—due to the shapes being unfamiliar and difficult to label—a 2-back version was utilised. Finally, due to the different number of stimuli used (i.e., 8 shapes versus 20 consonant letters), a different number of blocks was run, namely five blocks with six target shapes in each block—a total of 30 target shapes. Performance was determined by the overall proportion of correct responses. Testing time was approximately seven minutes. Five participants were excluded based on their performance in this task, as they had a mean response time of 3000 ms, indicating a lack of responding.

**Inhibition tasks.** *Stroop Task (Verbal)*. To measure verbal inhibition ability, a Stroop task was utilised [40]. In the task, participants were presented with a randomised sequence of colour words written in colour, one at a time, and were required to indicate the written colour with predefined key presses. Each colour word remained in the centre of the screen until the participant responded, allowing for response time to be measured; there was a 200 ms delay between words and a 400 ms error message for incorrect responses. The colours used were red ("D" key), green ("F" key), blue ("J" key), and black ("K" key). There were incongruent trials using colour words written in a different colour (e.g., "red" written in blue) or control trials that used coloured rectangles, with a total of 120 trials given (60 per condition). A practice block was given first that consisted of 18 trials (9 per condition). Inhibition ability was determined by the average difference in response time between correct incongruent and control trials, where a smaller difference indicated better inhibition ability. Testing time was approximately three minutes.

*Parametric Go-No-Go Task (Verbal)*. To measure verbal inhibition ability, we also used a Go-No-Go task [41] with letters as stimuli [42]. Generally, this task involved presenting a stream of letters to participants (for 500 ms each) with instructions to press the space bar for target letters. Target letters were defined differently throughout the task based on three different levels of difficulty: at Level 1, participants were instructed to respond to letters "r", "s", or "t"; at Level 2, the instruction was to respond to letters "r" and "s" but only when they were not repeats (i.e., if you respond to the letter "r", then do not respond to "r" again until after you have responded to the letter "s"); Level 3 was identical to Level 2 but used all three target letters ("r", "s", "t"). At each level, a short practice block of 20 (Levels 1 & 2) or 25 trials (Level 3) was given first, showing letters for 1000 ms each and instructing participants on when to respond, with corrective feedback. Only Level 3 data were used to assess inhibition ability due to the high aptitude of our sample. Level 3 had a total of 552 trials including 64 target ("go") trials (respond to "r", "s", or "t") and 26 lure ("no-go") trials (repeats of "r", "s", or "t"). Inhibition ability was determined by the proportion of correct responses to lure trials. Testing time was approximately seven minutes. Five participants were excluded due to below-chance target-trial performance.

*Arrow-Flanker Task (Non-verbal)*. To measure non-verbal inhibition ability, a Flanker task was utilised [43] with arrows as stimuli [44]. On each trial, a fixation cross appeared centrally for 1750 ms, followed by a row of five arrows. Participants indicated whether the central arrow pointed left ("Q" key) or right ("P" key). The presented arrows could either be congruent (i.e., all arrows pointing the same direction) or incongruent (i.e., central arrow pointing the opposite direction to the four surrounding arrows). If participants did not respond within 1750 ms, the correct answer was indicated on-screen, and the trial marked incorrect. The task started with eight practice trials, followed by a main block of 48 randomised trials, which started with a 3000 ms message that read "get ready". Inhibition ability was determined by the mean response time difference between correct congruent and incongruent trials, with a smaller difference indicating better ability. Testing took approximately three minutes.

**Shifting tasks.** *Number-Letter Task (Verbal)*. To measure verbal shifting ability, we used a Number-Letter task [19]. This task presented participants with a 2 × 2 matrix. In each trial, a number-letter pair (e.g., E8, 7H, etc.) was presented in one of the matrix cells, starting in the top-left cell and then moving in a clockwise fashion across trials. When the pair appeared in one of the top two quadrants, participants were required to indicate whether the letter was a consonant ("E" key) or a vowel ("I" key); when the pair appeared in one of the bottom two quadrants, participants classified the number as odd ("E" key) or even ("I" key). Interstimulus time was 150 ms for correct trials and 1500 ms for incorrect trials (with error feedback). Practice blocks consisting of 32 trials each were first given for the number and letter tasks

separately; this was followed by 16 practice trials for the combined task, with eight switch trials (i.e., switching from number to letter task or vice versa) and eight non-switch trials. The main block presented 128 trials, with 64 switch trials and 64 non-switch trials. Shifting ability was determined by the average response time difference between correct switch and non-switch trials, where a smaller difference meant better ability. Testing time was approximately seven minutes.

*Category-Switch Task (Verbal).* To measure verbal shifting ability, we also used the Category-Switch task [38,45]. In this task, participants were presented with a stream of nouns (from a pool of 16). Each noun was accompanied by a heart or cross symbol that specified the task to be performed; nouns with a heart were classified as living ('E" key) or non-living ("I" key) things; nouns with a cross were categorized as objects bigger ("E" key) or smaller ("I" key) than a basketball. The intertrial interval was 500 ms for correct responses and 1500 ms for incorrect responses (with corrective feedback). Practice blocks, each consisting of 32 trials, were given first for each individual task; this was followed by 16 combined-task practice trials (8 switch, 8 non-switch). The main block comprised 32 trials (16 switch, 16 non-switch). Shifting ability was determined by the mean response time difference between correct switch and non-switch trials, where a smaller difference indicated better ability. Testing time was approximately six minutes.

*Trails Task (Non-verbal).* To measure non-verbal shifting ability, we used a modified Trails task [46,47]. This task has two components called Trails A and Trails B. Both tasks involve an array of circles; in Trails A, the circles contain a sequence of numbers (i.e., 1–26), while in Trails B, half the circles contain a sequence of numbers (i.e., 1–13) and the other half a sequence of letters (i.e., A-M). Participants were given Trails A first, where the aim was to draw a line (with the computer mouse) through the numbered circles in the correct sequence (i.e., 1-2-3. . .26). Subsequently, Trails B was given, where the aim was to draw a line that alternated between numbered and lettered circles in the correct sequence (i.e., 1-A-2-B. . .13-M). If participants made an error, they were informed and instructed to continue from the last correct circle. Shifting ability was determined by performance on Trails B only, with a shorter completion time indicating greater ability. Testing time was approximately three minutes. Twenty-three participants were excluded based on performance on this task; specifically, participants whose error rates were greater than 2.4 interquartile ranges above the third quartile [48] on Trails A or B were excluded.

**CIE task.** This task was implemented using Qualtrics software [49]. Eight event reports and accompanying questionnaires were given in total. Each report contained a description of a different event and had a critical piece of information related to the event's cause, which subsequently was or was not retracted. There were four retraction-condition reports and four control-condition reports; conditions alternated, with a control report always given first, in order to avoid build-up of retraction expectations. Presentation order of the reports was counterbalanced using a Latin square (see Table S1 in the Online Supplement, available at https://osf.io/hw47f/). Questionnaires containing inference questions related to the event reports were subsequently given and followed the same order as the reports. An effort question designed to ascertain participant engagement was given at the end of the task.

*CIE Task Event Reports.* Each report consisted of two articles that were each around 100 words—following precedent [e.g., 11]. The first article always contained the critical information about the event's cause. The second article contained either a retraction of the critical information (retraction condition) or additional neutral information about the event (control condition). The no-retraction control condition was used as a baseline of participant's reliance on the critical information, and neutral information was given in place of the retraction so that report length was closely matched between conditions. For example, one report described an incidence of mass fish deaths in a river and suggested that the cause was chemical waste

dumping by a riverside pharmaceutical company; this was subsequently retracted in the second article. Minimum presentation time for the articles within each report was 15 seconds, such that participants could not continue to the next report until that time had elapsed. Once all reports were encoded, participants completed a one-minute distractor task—in line with previous studies [e.g., 17]. All reports are provided in the Online Supplement.

*CIE Task Questionnaires.* Similar to previous work [e.g., 18] questionnaires comprised one memory and four inference questions per report. The multiple-choice memory questions were provided to ensure that participants had adequately encoded the reports and targeted details unrelated to the critical information (e.g., "What contributed to low water storage levels in the affected region?"–a. drought; b. over-usage; c. containment leak; d. pump failure). Participants were excluded if they incorrectly answered more than 5 out of 8 basic memory questions (i.e., performance at chance level; $n = 5$), following previous studies [e.g., 17]. The inference questions were designed to measure reliance on the critical information. For each report, three inference questions asked participants to rate their endorsement of a statement using an 11-point Likert scale (e.g., "Chemical contamination contributed to the incident."–*strongly disagree* {0} to *strongly agree* {10}); one inference question used a multiple-choice format (e.g., "What do you think was the cause of the fish deaths?"–a. chemical spill; b. water temperature; c. virus; d. algal bloom; e. none of the above). All questionnaires are provided in the Online Supplement. Testing time was approximately 15 minutes.

*CIE Task Effort Question.* The effort question was a multiple-choice question, and was presented as follows: "*Before you go, please truthfully answer the following question: In your honest opinion, should we use your data? This is not related to how well you think you performed but whether you put in a reasonable effort. Please be assured that your response to this question will have no effect on your assignment of credit points. We just need to know what data to include in our analyses*". There were three possible responses, namely: (i) yes, I put in a reasonable effort; (ii) maybe, I was a little distracted; and (iii) no, I really was not paying attention. Participants selecting option 3 were excluded ($n = 3$), following precedent [e.g., 32].

*CIE Score Calculation.* Misinformation reliance was determined by using inference scores derived from responses to the inference questions. The 0–10 Likert-scale responses were divided by 10 to convert them to a 0–1 scale, while multiple-choice responses were coded as either 0 or 1. Responses were then averaged for each report, and four difference scores were calculated by pairing retraction and control report scores by order of magnitude (i.e., subtracting highest-to-lowest ranked control-report inference scores from highest-to-lowest ranked retraction-report scores); note that the specific way in which the retraction and control conditions were paired was inconsequential to results and only influenced internal-consistency reliability estimates. These four difference scores then formed the observed variables that were used as indicators of the CIE latent variable in our SEM analysis. Finally, the four difference scores were used to calculate a single mean score, which served as the observed CIE score; while we acknowledge that this score does not reflect the CIE per se, but rather reflects retraction efficacy, to stay consistent with previous research [e.g., 17,18] we refer to it as a CIE score.

## Procedure

Presentation order for the executive function tasks was: 1st, Trails; 2nd, Go-No-Go; 3rd, Letter *N*-Back; 4th, Number Letter; 5th, Stroop; 6th, Shape *N*-Back; 7th, Category Switch; 8th, Flanker; 9th, Keep Track. This presentation order ensured that tasks of the same type (i.e., updating, inhibition or shifting) were separated, so as to reduce practice effects. Participants then completed the CIE task. Task instructions were given on-screen. In total, the experiment took approximately 60 min to complete.

## Data analysis

Model fit of our EF measurement models was evaluated with CFA, while SEM was used to determine the relationship between EF subcomponents and CIE susceptibility. For our CFAs, the analysis plan was to first assess the model fit of Miyake et al.'s [19] original correlated three-factor model, wherein updating, inhibition, and shifting each form latent variables that are inter-correlated. Failing acceptable model fit, the plan was to next test Miyake et al.'s [30] alternative nested bi-factor model, which has specific updating and shifting factors but no inhibition factor—inhibition is instead subsumed in a general EF factor that loads onto all tasks. In case of unacceptable model fit, we then planned to test alternative models reported in previous literature. For our SEM analysis, we took the EF measurement model found to fit our data, and then regressed our CIE latent variable onto the EF-subcomponent latent variables of said model.

All CFAs and SEMs were run in AMOS 27 [50] using maximum-likelihood estimation. We used standardisation as the scaling method for our latent variables. Our point-estimate CIs were estimated using bootstrapping (2,000 samples). Model fit was determined using the following criteria from Schweizer [51]: comparative fit index (CFI) $\geq$ .950; Tucker-Lewis index (TLI) $\geq$ .950; standardised root mean-square residual (SRMR) $<$ .08; root mean square error of approximation (RMSEA) $<$ .06 (including 90% CIs). We report implied model $\chi^2$ statistics for completeness. Any necessary model comparisons in our CFAs and SEM analysis were based on the following criteria: TLI difference $>$ .010 [52]; Bayesian information criterion (BIC) difference $>$ 2.00 ([53]; lower BIC values indicate better fit); and Akaike information criterion (AIC) difference $>$ 2.00.([54]; lower AIC values indicate better fit) Observed-score correlation effect sizes were based on criteria established by Gignac and Szodorai ([55]; small, $r \geq$ .10; typical, $r \geq$ .20; large, $r \geq$ .30), as were effect sizes for true-score (latent-variable) correlations (small, $r \geq$ .15; typical, $r \geq$ .25; large, $r \geq$ .35). Finally, before conducting our analyses, any extreme values in our response-time data that were indicative of invalid responding (i.e., $<$ 300 ms [$<$ 200 ms for the Flanker task] or $>$ 5 s) were winsorised to the next non-conspicuous value.

## Results

### Internal-consistency reliability and descriptive statistics

Before running preliminary analyses, we estimated internal-consistency reliability. As can be seen in Table 1, most task scores yielded internal reliability greater than .70, with the exception of the Keep-Track, Stroop, Category-Switch, and CIE tasks. Furthermore, skew and kurtosis were within acceptable ranges for all tasks except the Flanker task (skew $<$ |2| & kurtosis $<$ |9|; [56,57]). The deviant kurtosis of the Flanker task was likely due to an outlier, however, we employed bootstrapping in our analyses for robustness against deviations to normality.

### Preliminary analyses

Prior to testing the relationship between the EF-subcomponent latent variables and the CIE latent variable, we (i) performed a manipulation check on the CIE task to establish the presence of a CIE by determining if there was a statistically significant difference between the retraction and control conditions, and whether retraction condition scores were statistically different from zero (following precedent; [e.g., 11]); (ii) performed a correlation analysis on all tasks; and (iii) performed CFAs to test which model best fitted our EF data.

The manipulation check on the CIE task confirmed that there was a significant difference between retraction ($M$ = .31, $SD$ = .17) and control ($M$ = .51, $SD$ = .11) conditions in the

**Table 1. Descriptive statistics and internal-consistency reliability for EF tasks and CIE task (N = 255).**

| Task (units) | M | SD | Range | Skew | Kurt | Reliability |
|---|---|---|---|---|---|---|
| Letter N-Back (%) | .83 | .08 | .63–1.00 | .10 | -.37 | .75[a] |
| Keep-Track (%) | .69 | .12 | .08 –.97 | -1.11 | 3.99 | .61[a] |
| Shape N-Back (%) | .81 | .10 | .40 –.99 | -.82 | .78 | .85[a] |
| Stroop (ms)[b] | -129.7 | 101.1 | -600.6–104.9 | -1.06 | 1.96 | .39[ac] |
| Go-No-Go (%) | .65 | .16 | .12 –.96 | -.37 | .12 | .70[a] |
| Arrow Flanker (ms)[b] | -30.1 | 32.6 | -237.1–94.5 | -1.48 | 10.16 | .81[ac] |
| Number-Letter (ms)[b] | -715.2 | 348.0 | -1803.0–72.7 | -.77 | .35 | .95[ac] |
| Category-Switch (ms)[b] | -290.4 | 218.6 | -1172.9–459.2 | -.92 | 2.52 | .36[ac] |
| Trails (s)[b] | -59.97 | 19.58 | -152.5 –-27.1 | -1.41 | 3.14 | N/A |
| CIE Event Memory | 6.15 | 1.39 | 3–8 | -.45 | .57 | * |
| CIE | -.21 | .19 | -.73 –.41 | .03 | -.09 | .57[a] |

*Note*. Kurt, Kurtosis; %, proportion correct; [a] McDonald's ω. [b] All response-time-based task scores were multiplied by -1 to make correlations between tasks easier to interpret. [c] The difference-score reliability formula was used to correct these estimates. * McDonald's ω could not be calculated due to negative covariances between items.

appropriate direction, $t(254) = 17.24$, $p < .001$, $d = 1.08$. Furthermore, in checking for the presence of a CIE with one-sample $t$-tests, we found a significant difference between all retraction condition scores and zero, $t(254) \geq 15.14$, $p < .001$, $d \geq .95$, indicating the presence of a CIE.

As can be seen in Table 2, the updating tasks (i.e., Shape *N*-Back, Keep-Track, and Letter *N*-Back tasks) demonstrated large, positive correlations, while the switching tasks (i.e., Number-Letter, Category-Switch, and Trails tasks) demonstrated mostly typical, positive correlations; however, note that the Trails task correlated more highly with the updating tasks than with the other switching tasks. As for the inhibition tasks, the Stroop and Go-No-Go tasks showed a small, positive correlation, while the Flanker and Go-No-Go tasks had a small, negative correlation; the Stroop and Flanker tasks did not correlate significantly. The event-related-memory subtask from the CIE task demonstrated typical, positive correlations with both N-Back tasks and the Go-No-Go task, as well as small, positive correlations with the Keep-Track, Stroop,

**Table 2. Correlations between the Nine EF tasks and CIE task (N = 255).**

| | 1) | 2) | 3) | 4) | 5) | 6) | 7) | 8) | 9) | 10) | 11) |
|---|---|---|---|---|---|---|---|---|---|---|---|
| 1) N-Back L | - | | | | | | | | | | |
| 2) Keep-Track | .30** | - | | | | | | | | | |
| 3) N-Back S | .49** | .39** | - | | | | | | | | |
| 4) Stroop | .18** | .25** | .20** | - | | | | | | | |
| 5) GnG | .24** | .23** | .30** | .14* | - | | | | | | |
| 6) Flanker | .02 | -.00 | .07 | .03 | -.15* | - | | | | | |
| 7) Num-Lett | .01 | .05 | .10 | .01 | -.04 | -.06 | - | | | | |
| 8) Cat-Swi | .01 | .02 | .06 | -.01 | -.01 | -.07 | .29** | - | | | |
| 9) Trails | .33** | .25** | .32** | .14* | .14* | -.01 | .21** | .16* | - | | |
| 10) Ev-Mem | .24** | .17** | .26** | .18** | .20** | .04 | .10 | .09 | .19** | - | |
| 11) CIE | -.28** | -.19** | -.24** | -.05 | -.18** | -.03 | -.06 | -.07 | -.20** | -.46** | - |

*Note*. L, Letter; S, Shape; GnG, Go-No-Go; Num-Lett, Number-Letter; Cat-Swi, Category-Switch; Ev-Mem, CIE Event-Memory

*, < .05

**, < .01.

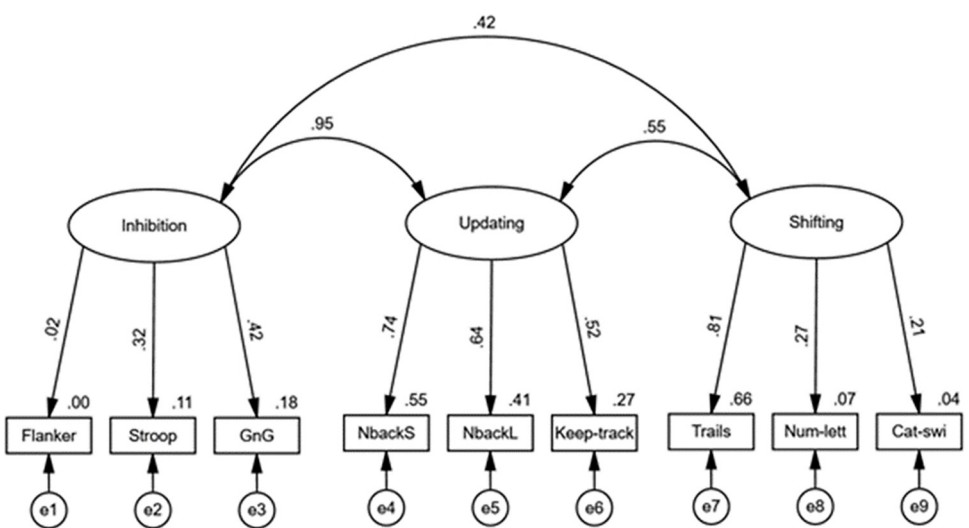

**Fig 1. Confirmatory factor analysis of EF-subcomponent latent variables in correlated three-factor model.** Note. Straight lines with single arrows are regression paths. The curved lines with double arrows are correlations. The observed variables at the bottom represent EF-task scores. Numbers on top-right of observed variables represent proportion of variance accounted for in each variable. Error terms associated with each observed variable are indicated by e1-9. Factor loadings and correlations are fully standardised.

and Trails tasks. Finally, the CIE task demonstrated mostly typical, negative correlations with the three updating tasks, the Go-No-Go task, and the Trails task, as well as a large, positive correlation with the event-related-memory subtask.

A CFA conducted on Miyake et al.'s [19] correlated three-factor model of the EF-subcomponent latent variables yielded unacceptable model fit, based on the incremental close-fit indices, $\chi^2(24) = 41.21$, $p = .016$, $CFI = .925$, $TLI = .887$, $SRMR = .055$, $RMSEA = .053$ (90% CI [.023, .080]). Fig 1 shows that almost all factor loadings were positive and significant (all $ps <$ .008) except for the Flanker-task loading ($p = .825$). There was also a statistically significant, large, positive correlation between the updating and shifting latent variables ($r = .55$, 95% CI [.25, .76], $p < .001$) and a statistically significant, large, positive correlation between the shifting and inhibition latent variables ($r = .42$, 95% CI [.06, 1.99], $p = .019$). Moreover, and notably, the correlation between the updating and inhibition latent variables was large, positive and statistically significant ($r = .95$, 95% CI [.41, 3.63], $p < .001$). The almost perfect inhibition-updating correlation, with the upper 95% CI exceeding 1—which was also true of the inhibition-shifting correlation—implied a lack of factor distinctness associated with at least one dimension. Furthermore, the stronger loadings on the updating factor, versus the inhibition factor, implied that the construct measured by said factors was updating, suggesting inhibition failed to be measured in a construct-valid manner. Given this, we decided to remove the inhibition tasks from further analyses. Therefore, we did not test Miyake et al.'s [30] alternative nested bi-factor model as planned, and instead tested a correlated two-factor model with updating and shifting factors.

In our CFA of a correlated two-factor model defined by updating and shifting latent variables, model fit was found to be unacceptable, $\chi^2(8) = 22.90$, $p = .003$, $CFI = .919$, $TLI = .847$, $SRMR = .060$, $RMSEA = .086$ (90% CI [.046, .128]); however, when we allowed a cross-loading for the Trails task, model fit was excellent, $\chi^2(7) = 4.69$, $p = .698$, $CFI = 1.000$, $TLI = 1.027$, $SRMR = .020$, $RMSEA = .000$ (90% CI [.000, .059]). Indeed, the Trails task has been found to yield cross-loadings onto updating and shifting factors previously [e.g., 58]. As can be seen in Fig 2, all factor loadings were positive for both latent variables, and all loadings were significant

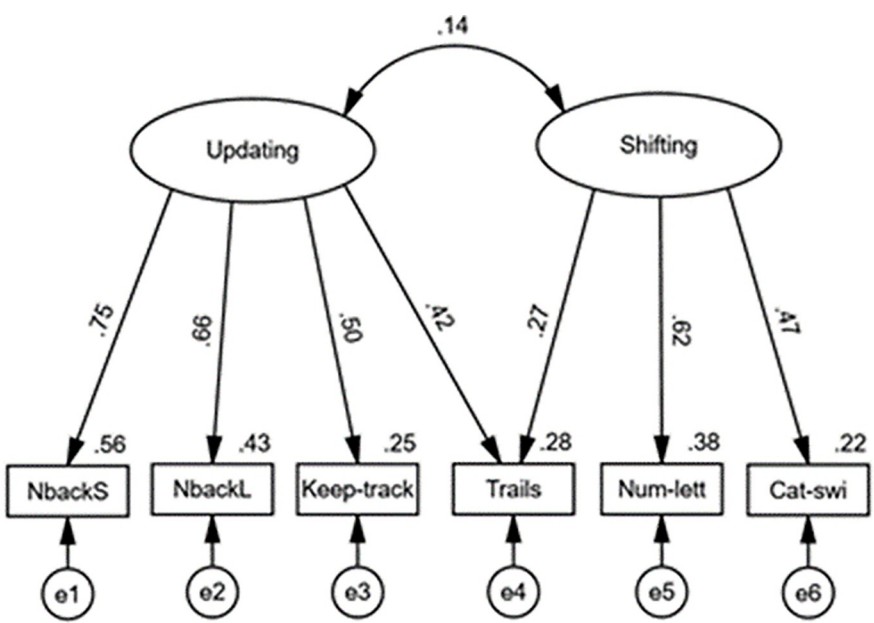

**Fig 2. Confirmatory factor analysis of EF-subcomponent latent variables in correlated two-factor model.** Note. Straight lines with single arrows are regression paths. The observed variables at the bottom represent EF-task scores. Numbers on top-right of observed variables represent proportion of variance accounted for in each variable. Error terms associated with each observed variable are indicated by e1-6. Factor loadings and correlations are fully standardised.

($p < .001$). Further, while the latent variable correlation was significant and large in the standard correlated two-factor model ($r = .55$, 95% CI = [.27,.76], $p < .001$), it was non-significant with the cross-loading included ($r = .14$, 95% CI [-.08, .35], $p = .208$). Note that including the Trails cross-loading in the correlated three-factor model also produced acceptable model fit; however, due to issues highlighted for the inhibition factor in the above paragraph, the correlated two-factor model was preferred. For transparency, results of analyses with the correlated three-factor model have been provided in the Online Supplement.

### Structural Equation Modelling (SEM) of EF and CIE

Following the identification of a well-fitting measurement model for our EF data, and to test the hypothesised relationship between EF and the CIE, we conducted a SEM wherein the CIE latent variable was regressed onto the EF-task latent variables (updating and shifting). The model demonstrated excellent model fit, $\chi^2(31) = 38.36$, $p = .170$, $CFI = .975$, $TLI = .964$, $SRMR = .042$, $RMSEA = .031$ (90% $CI = [.000, .059]$); further, as shown in Fig 3, the beta weight associated with the updating and CIE latent variables was negative ($\beta = -.54$, 95% $CI$ [-.70,-.37]) and statistically significant ($p < .001$), while the regression between the shifting and CIE latent variables was negative and non-significant ($\beta = -.16$, 95% $CI$ [-.36, .08], $p = .245$). A total of 33% (95% $CI$ [17, 52%]) of the CIE's true score variance was accounted for by the model. This supported our hypothesis and suggested that individual differences in EF, particularly working-memory updating, were predictive of individual differences in the CIE (i.e., higher EF ability predicts lower CIE susceptibility). Finally, while a significant negative correlation was found between event-related memory and CIE susceptibility, we were unable to include event-related memory in our SEM analyses due to an unacceptably low KMO value for its measure ($KMO = .56$; [59]).

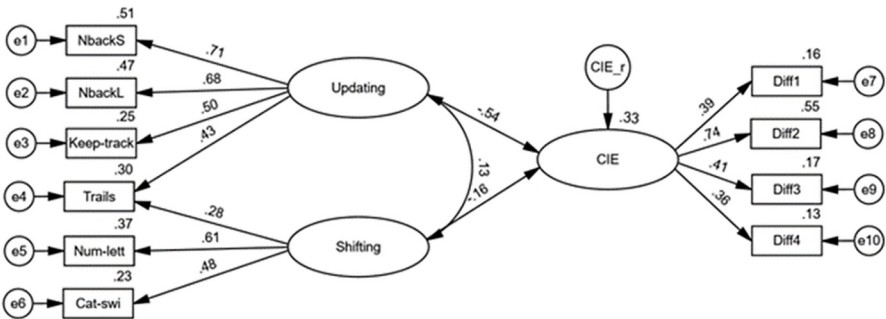

**Fig 3. Structural equation model of multiple regression between EF-subcomponent latent variables and CIE latent variable.** Note. Straight lines with single arrows are regression paths. The observed variables on the left represent EF-task scores. The observed variables on the right represent CIE scores. Numbers on top-right of observed variables and CIE latent variable represent proportion of variance accounted for in each variable. Error terms associated with each observed variable are indicated by e1-10. Factor loadings and correlations are fully standardised.

## Supplementary factor analysis

Given that results of latent variable analyses can be unstable at sample sizes under 400 with the factor-loading magnitudes observed in our model [60], we performed a supplementary, unrestricted factor analysis of our two EF factors (i.e., updating and shifting) to help confirm the veracity of findings associated with our latent-variable analyses. Specifically, we factor-analysed the six indicators associated with the updating and shifting dimensions, as well as the CIE observed scores. The supplementary factor analysis was run using the *EFAutilities* package in R. Maximum likelihood estimation was utilised with direct oblimin rotation and Kaiser normalisation, while bootstrapped 95% CIs (2,000 samples) were calculated for factor loadings and the factor correlation; this served as a measure of statistical significance for both.

Firstly, the KMO value suggested that our sample was suitable for factor analysis ($KMO = .72$; [59]) as did the result of Bartlett's test of sphericity, $\chi^2 = 224.38$, $p < .001$. As shown in Table 3, the extracted communalities ranged from .14 to .51. As further seen in Table 3, the pattern of loadings replicated the findings of our CFAs; that is, the updating tasks (i.e., *N*-Back and Keep-Track tasks) loaded significantly onto an updating factor, the shifting tasks (i.e., Number-Letter, Category-Switch, and Trails tasks) loaded significantly onto a shifting factor, and the Trails task cross-loaded. Moreover, the CIE task loaded negatively and significantly onto the updating factor, $\lambda = -.37$, 95% CI [-.50, -.24], but not the shifting factor, $\lambda = -.03$, 95%

**Table 3. Extracted communalities and fully standardized factor loadings (with 95% CI) for our two extracted factors that contained the updating tasks, shifting tasks and CIE task.**

| Task | Extracted $h^2$ | Updating | | Shifting | |
|---|---|---|---|---|---|
| | | $\lambda$ | 95% CI | $\lambda$ | 95% CI |
| N-Back L | .48 | .71 | .58 –.84 | -.13 | -.28 –.02 |
| Keep-Track | .25 | .50 | .38 –.62 | -.02 | -.17 –.12 |
| N-Back S | .51 | .72 | .60 –.85 | -.02 | -.14 –.12 |
| Num-Lett | .39 | .00 | -.13 –.14 | .63 | .23–1.02 |
| Cat-Swi | .23 | -.01 | -.12 –.10 | .48 | .12 –.83 |
| Trails | .29 | .44 | .29 –.58 | .23 | .03 –.43 |
| CIE | .14 | -.37 | -.50 –-.24 | -.03 | -.19 –.14 |

*Note.* $h^2$, communality; $\lambda$, factor loading; CI, confidence interval; L, Letter; S, Shape; lett, Letter; swi, Switch.

CI [-.19, .14], thus corroborating the results of our SEM analysis. Finally, the correlation between the updating and shifting factors was $r = .22$, which was significant based on the 95% CIs [.03, .39]; this provides further supporting evidence that a correlated two-factor model was appropriate for our EF data.

## Discussion

The aim of the current, exploratory, study was to investigate whether executive function (EF) could predict CIE susceptibility. It was hypothesised that greater EF ability would predict lower CIE susceptibility; we thus predicted that (i) there would be significant negative correlations between one or more EF tasks and the CIE task, and (ii) there would be a significant negative beta weight between one or more EF latent variable(s) and the CIE latent variable. This hypothesis and its related predictions were supported by the present results.

Our CIE task having a negative correlation of typical effect size [55] with our updating, Go-No-Go, and Trails tasks provides initial evidence that greater EF ability is associated with lower CIE susceptibility. However, given that the Go-No-Go and Trails tasks also had typical to large-sized positive correlations with all three updating tasks, the correlation analysis suggests that updating ability specifically relates to CIE susceptibility. This suggestion was confirmed by the significant, negative, standardised beta weight found between our updating and CIE latent variables ($β = -.54$), which demonstrated that updating ability can predict CIE susceptibility. Furthermore, our EF model explained 33% of the variance in the CIE latent variable, a large proportion of which was due to the updating factor. Moreover, an unrestricted factor analysis of our EF model and CIE data—conducted as a supplementary post-hoc analysis—showed that CIE-task scores loaded negatively with the updating-task scores, suggesting that the CIE task was tapping into updating ability. This consequently suggests that updating ability is an intrinsic aspect of CIE susceptibility. Thus, overall, the current results provide preliminary evidence that EF is an important determinant of people's susceptibility to the CIE, and that working-memory updating may be particularly crucial.

The current results expand upon the findings of Brydges et al. [17], who demonstrated a predictive relationship between working-memory capacity and CIE susceptibility. However, it should be noted that Sanderson et al. [18] failed to replicate Brydges et al.'s findings; they also conducted a reanalysis of Brydges et al.'s data that yielded a non-significant relationship between the CIE and working-memory-capacity variables. Yet, these contradictory results may have been due to the working-memory capacity measures used by Brydges et al. and Sanderson et al.; that is, these measures may not have tapped into working-memory updating sufficiently to produce a reliable effect with CIE measures. Nonetheless, Brydges and colleagues speculated that their results could be evidence for the mental-model-updating account of the CIE, as the updating mechanism central to this account arguably relies on working memory. This speculation aligns with the present study's results. Specifically, the present study showed that individual differences in working-memory updating significantly predicted individual differences in the CIE, with better updating predicting lower CIE susceptibility. This predictive relationship was also not limited to the verbal domain, like the working-memory-capacity relationship found in Brydges et al. [17] was. Thus, people's ability to update representations in their working memory appears to be linked to their ability to discount corrected information in reasoning. Consequently, the current study provides individual-differences evidence to support the mental-model-updating account of the CIE [for a recent review, 61].

However, it should be acknowledged that the updating measures used in the current study may not have been pure measures of updating; that is, other aspects of working memory may have also been measured. Indeed, while the Keep-track and N-Back tasks are commonly used

as updating measures [e.g., 19,38,62,63], some evidence suggests that these tasks also measure more basic storage and maintenance operations in working memory [e.g., 64,65]. Therefore, despite our attempts to isolate updating-specific variance with a latent-variable approach, we cannot definitively conclude from our results that updating was the only working-memory process predicting CIE susceptibility.

Of course, the current results do not preclude the possibility that other cognitive factors may determine CIE susceptibility. Most immediately, the current study proposed a potential predictive relationship between prepotent-response inhibition and the CIE based on the selective-retrieval account of the CIE [e.g., 5,12]. To recap, the selective-retrieval account suggests that a CIE may arise if misinformation is selectively retrieved (e.g., based on automatic familiarity processes) and misinformation-based responses are not inhibited at test, with such inhibition potentially facilitated by strategic recollection of a relevant correction. Thus, better ability to inhibit prepotent responses may reduce CIE occurrence. However, we were unable to assess this proposed relationship, as our inhibition factor did not converge.

While we suggest that future research reattempt a latent-variable analysis of the potential inhibition-CIE relationship, it must be acknowledged that measurement of the inhibition construct has generally proven problematic [e.g., 66–68]. In fact, it is even debated whether inhibition can be measured as a unitary construct, with some research suggesting that inhibition may be composed of separable but related subfactors [69–72]. However, attempts to measure these inhibition subfactors have also yielded inconsistent outcomes, as demonstrated with prepotent-response inhibition in the current study and other studies using Miyake's EF model [see 29 and 31 for reviews]. Friedman and Miyake [31] indeed stated that many failures to replicate their EF model have resulted from issues with the (prepotent-response) inhibition factor. Perhaps a potential reason why an inhibition factor was not found in the current study, and previous studies, lies in the measures used; more specifically, evidence suggests that experimental tasks designed to create a between-subjects effect (e.g., the Stroop effect in the Stroop task) can be unreliable at producing individual differences [e.g., 73]. Therefore, future studies investigating the relationship between inhibition and CIE susceptibility should carefully consider how inhibition is measured, consult past EF research that used a latent-variable approach, and select appropriate measures for forming an inhibition factor.

Beyond inhibition, verbal intelligence has been shown to influence susceptibility to the CIE [27]. Considering this, and evidence that executive function, particularly working-memory updating, correlates with intelligence [e.g., 38,75–76], it may be important to address whether the present results are separable from the intelligence-CIE relationship. More specifically, verbal intelligence may partially or completely mediate the predictive relationship found here between EF and CIE susceptibility, or vice-versa. Therefore, follow-up studies using a latent-variable approach may wish to conduct a mediation analysis with measures of verbal intelligence, EF ability, and CIE susceptibility. Such studies would help to elucidate whether the influence of executive processes on CIE susceptibility is separable from the influence of verbal cognitive ability.

Furthermore, Sanderson et al. [18] suggested that greater fidelity of the episodic memory representation of event reports predicted lower susceptibility to the CIE. Similarly, our analyses demonstrated a negative correlation between event-related memory and CIE susceptibility, though poor psychometric properties of our event-related-memory measure prevented further analysis with structural equation modelling. It may be worth further investigating whether episodic-memory ability more generally predicts CIE susceptibility, using more general measures of episodic memory. Moreover, it is possible that episodic-memory abilities could interact with the relationship between EF and the CIE. Indeed, those with better episodic memory may generate higher-fidelity mental models, which may in turn be easier to update. Future

investigations could thus seek to assess how the relationship between EF and CIE susceptibility may change across the spectrum of episodic-memory ability, or in other words, determine if episodic memory moderates the EF-CIE relationship.

Regarding the measurement of Miyake's model more specifically, it is notable that our cor-related two-factor (updating and shifting) model, while not replicating Miyake's original or alternative model, does have precedence [58,77]. Hull et al. [77] found a correlated two-factor model in line with the current study. Van der Sluis et al. [58] found a nested two-factor model with updating and shifting factors nested within a "naming" factor that comprised non-EF aspects (i.e., baselines) of each EF measure (e.g., congruent condition in Stroop task, trails A in Trails task, etc.). Interestingly, both studies and the current study tested different age groups, namely primary-school children [77], older adults [58], and undergraduate students (current study). Thus, results across all three studies could provide tentative evidence that a two-factor updating-and-shifting EF model can manifest across the lifespan. However, we must stress that this goes against the larger evidence [see 29 for a meta-analytic review] and so we cannot provide definitive conclusions regarding EF's underlying structure.

Also notable was our replication of van der Sluis et al.'s [58] finding that Trails B cross-loaded onto updating and shifting factors. The Trails task has traditionally been considered a measure of shifting [47]; this makes sense, as participants who complete Trails-B must actively switch between number and letter sequences as quickly as possible to perform well. However, it is conceivable that performance on Trails B is also linked to updating ability. Specifically, one could argue that participants in Trails B must update letter and number sequences as well as switch between them; thus, greater updating ability would reduce time spent updating each sequence between switches, reducing overall completion time and error rates. However, fur-ther psychometric investigation will be required to confirm whether such conjecture is supportable.

Regarding measurement of the CIE, the current study provides further psychometric data on the CIE paradigm, which has been hitherto limited.[78] In previous individual-differences research using the CIE paradigm [e.g., 17,18,32] the reported estimates of internal-consistency reliability have been problematic. Brydges et al. [17] and Sanderson et al. [18] reported esti-mates of $\alpha = .65$ and $\alpha = .46$, respectively. Similarly, McIlhiney et al. [32], who used two parallel CIE tasks in a test-retest format, reported estimates of $\omega_{time1} = .53$ and $\omega_{time2} = .60$. While these findings could suggest that the CIE paradigm suffers from similar issues found when using other experimental tasks in individual-differences research [e.g., 73], McIlhiney et al.'s [32] findings suggested that the CIE paradigm showed acceptable stability in individual-differ-ences variation. Following recommendations cited by McIlhiney et al., we attempted to allevi-ate reliability issues by incorporating additional retraction and control items; however, our estimated internal consistency of $\omega = .57$ indicated no improvement to reliability. Given this result, we cannot recommend that future studies increase item count in the CIE task. While one could argue that adding further items to retraction and control conditions is needed, we would counterargue that doing so may introduce memory effects that could obscure CIE mea-surement; that is, a longer task with more reports to remember may overtax people's memory capacity, making it harder for the researcher to distinguish between forgetting and genuine misinformation reliance. It should be noted, however, that our CIE task correlated well with other tasks, therefore the limited internal consistency did not appear to limit our study exces-sively. Nevertheless, we do recommend that future research continues to investigate ways of improving the psychometric properties of the CIE paradigm.

In practical terms, the results of the current study could be useful to inform attempts to address the spread of misinformation, such as the intervention strategies summarised in The Debunking Handbook 2020 [79]. While information-focused interventions have

demonstrated efficacy in the lab, the efficacy of such interventions should be tested on those with lower or compromised cognitive abilities (e.g., those with compromised EF). However, it should be acknowledged that individual differences in executive function will only play a small role in real-world CIE examples. Furthermore, given that cognitive abilities can be difficult to change, intervention strategies focused on inoculating and educating against misinformation influence may help to support information-focused interventions—particularly given their generally demonstrated efficacy [e.g., 80–91].

Apart from the already-identified issues with CIE task reliability and the potential mediating role of intelligence, one limitation that should be acknowledged is range restriction due to our undergraduate-only sample—especially since age differences in EF ability have been found across the subcomponents we tested [e.g., 29,67,92–100]. Due to this range restriction, the effects reported here are likely underestimates, and so future research should seek to replicate our findings in a more heterogenous sample.

## Summary and conclusion

To summarise, the current study provides evidence that executive function, particularly working-memory updating, can play a significant role in determining susceptibility to the continued influence of misinformation. Unfortunately, this means that those with lowered executive abilities, particularly in the domain of working-memory updating, are at higher risk of misinformation's influence. This carries particular implications for those in our society with impaired executive ability.(e.g., older adults; [93]) It is hoped, then, that our findings will support further development of real-world intervention strategies designed to combat the effects of misinformation.

## Acknowledgments

There are no acknowledgements to be made.

## Author Contributions

**Conceptualization:** Paul McIlhiney, Gilles E. Gignac, Ullrich K. H. Ecker, Michael Weinborn.

**Data curation:** Paul McIlhiney.

**Formal analysis:** Paul McIlhiney, Gilles E. Gignac.

**Investigation:** Paul McIlhiney.

**Methodology:** Paul McIlhiney, Gilles E. Gignac, Ullrich K. H. Ecker, Briana L. Kennedy, Michael Weinborn.

**Project administration:** Paul McIlhiney, Gilles E. Gignac, Ullrich K. H. Ecker, Michael Weinborn.

**Software:** Paul McIlhiney.

**Supervision:** Ullrich K. H. Ecker, Michael Weinborn.

**Validation:** Paul McIlhiney, Gilles E. Gignac.

**Visualization:** Paul McIlhiney.

**Writing – original draft:** Paul McIlhiney.

**Writing – review & editing:** Paul McIlhiney, Gilles E. Gignac, Ullrich K. H. Ecker, Briana L. Kennedy, Michael Weinborn.

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
