## [Decision Letter · Decision Letter 0]

2 Aug 2022

PONE-D-22-17797Executive function and the continued influence of misinformation: A latent-variable analysisPLOS ONE

Dear Dr. McIlhiney,

Thank you for submitting your manuscript to PLOS ONE. I am sorry for the relative delay in processing the article - it has been a it of a challenge to find suitable reviewers, particularly now that it was close to the end of the semester and start of vacation period. I was however lucky to find two excellent expert reviewers that are quite knowledgeable on working memory, lay cognition and misinformation, and confirmatory factor analyses. I want to take this opportunity to thank the reviewers for their valuable input and careful suggestions for improvements. The reviewers see merit in your paper and believe that after careful revision the paper will make a fine contribution to the literature. This agrees with my own reading of the paper. Hence I am inviting a major revision. Please take in consideration of all the reviewer's comments, and respond to all of them to the best of your abilities. They provide complementary suggestions that I believe will improve the reading and robustness of the analyses and discussion points in the paper. In addition to the reviewer's comments, a few remarks arouse on my own reading of the paper that I would like to see addressed as well.First, I was a bit confused about the presentation of the results of Sanderson et al. vs. Brydges et al. It is not really clear what could explain their divergent results, and how the present study can help us further understand the relation between working memory and CIE. Please try to make a more straight point regarding the current state of the literature and what could explain these divergent findings, and how the present study advances the state of this debate.Second, as also noticed by Reviewer 1, there seems to be a conflation between updating and working memory capacity in the paper. Although R1 points it out only as a matter of measurement, I also noticed that the authors use these terms quite interchangeably, creating a bit of confusion, specially in describing their prior work (Sanderson et al. vs. Brydges et al.). Please be consistent in using the term updating when referring to scores derived from updating tasks, and working memory capacity to refer to scores of tasks that do not include updating (or that have a mix of both types of tasks). If the authors do not believe this distinction is relevant, I'd suggest that they are upfront about this in the intro (following the suggestions of R1) and opt for only one term throughout.Third, for the stroop task the authors say at some point that they had a congruent condition. Yet, it seems to me that the implemented condition would be more reasonably classified as a neutral baseline since no semantic information was provided (colored squares?).Fourth, I commend the authors for sharing their data, but as also noticed by R1, only processed data was provided. I wonder if the authors could also provide raw tasks scores. This would allow for more freedom for future studies to explore different task scoring methods. Please also consider adding a codebook  (i.e., meta-data file explaining the variables) to the project page.Fifth, I wondered a bit about the choice of building difference scores to create the CIE latent variable, and the rather arbitrary choice of which scores to subtract from which scores. I wonder why the authors did not create a CIE factor from a latent difference score? One could create a latent variable that reflects performance in all eight indicators and then one CIE factor that contains only the additional variance from the four correction conditions. Wouldn't that be less arbitrary? Here also providing the raw CIE task scores would be helpful, since at the moment, only the difference scores selected by the authors are available at the OSF and alternative model implementations are not possible.Sixth, I wondered about the memory measures from the CIE tasks. Could that data also be provided in the OSF? Is there a reason why at least the descriptives are not provided? Could this data provide any additional information?

We look forward to receiving your revised manuscript.

Kind regards,

Alessandra S. Souza, Ph.D.

Academic Editor

PLOS ONE

Journal Requirements:

2. Please provide additional details regarding participant consent. In the Methods section, please ensure that you have specified (1) whether consent was informed and (2) what type you obtained (for instance, written or verbal). If your study included minors, state whether you obtained consent from parents or guardians. If the need for consent was waived by the ethics committee, please include this information. 

*Please change "female” or "male" to "woman” or "man" as appropriate, when used as a noun (see for instance https://apastyle.apa.org/style-grammar-guidelines/bias-free-language/gender).

"PM’s contribution to this research was supported by an Australian Government Research Training Program (RTP) Scholarship. UKHE was supported by grant FT190100708 from the Australian Research Council."

"PM’s contribution to this research was supported by an Australian Government Research Training Program (RTP) Scholarship. 

UKHE was supported by grant FT190100708 from the Australian Research Council.

Reviewers' comments:

Reviewer's Responses to Questions

**Comments to the Author**

1. Is the manuscript technically sound, and do the data support the conclusions?

Reviewer #1: Yes

Reviewer #2: Yes

2. Has the statistical analysis been performed appropriately and rigorously? 

Reviewer #1: Yes

Reviewer #2: Yes

3. Have the authors made all data underlying the findings in their manuscript fully available?

Reviewer #1: Yes

Reviewer #2: Yes

4. Is the manuscript presented in an intelligible fashion and written in standard English?

Reviewer #1: Yes

Reviewer #2: Yes

5. Review Comments to the Author

Reviewer #1: The presented manuscript discussed if individual differences in the continued influence effect (CIE) of misinformation can be explained by executive functions. The theoretical introduction is clear, and the tested hypothesis are coherently derived from previous literature. The collected measures are well described and mostly adequate to test the hypothesis (see my comments below for specific critique points). The analysis is also clearly written and fitting for testing the hypothesis, although some more details could have been provided in the manuscript (see my detailed comments). The results are clearly presented and easy to comprehend. Likewise, the discussion summarizes the results well and integrates the presented findings well into the already existing literature. All in all, I think this is an interesting and well written manuscript that will make an important contribution for understanding the continued influence effect and individual differences therein. Yet, there are still some issues that I would like to see addressed prior to publication. These are outlined below.

1. Finding psychometrically good and valid indicators of executive functions is difficult and all in all I think the authors have made a balanced selection for the different executive functions. The only issue I see is that for updating only average performance in working memory tasks involving updating was used. This measure likely conflates individual differences in storage processes of working memory with executive processes like updating. Specifically, previous research has shown that updating tasks are equally good indicators of working memory capacity as simple binding tasks that do not require any updating and removal of outdated information (Wilhelm et al., 2013). This indicates that updating performance likely mainly measures individual differences in more basic WM processes not related to updating. This was further validated in a recent study of my own (Frischkorn et al., 2022). This is not to say, that I want the authors to cite my work, but I rather want to point out that a more nuanced discussion of their updating measurement is required, especially with respect to the discussion and implications of their results. It could be that primarily basic working memory processes such as storage & maintenance are related to individual differences in CIE, because the current measure does not sufficiently separate processes specific to updating from these basic processes.

2. I thank the authors for making their materials and data available. I was able to reproduce all the results and I think this is excellent for the promotion of open science. Yet, in reproducing the results I noted that some data analytical choices were not sufficiently reported in the manuscript.

a. How was the scaling of latent variables achieved? Did the authors fix the loading of one of the indicators to one (i.e., normalization), or did they fix the variance of the latent variables to one (i.e., standardization). This is a minor point, but for a complete description of the analysis please add this information

b. It seems that the authors have standardized the observed variables prior to estimating SEM. I understand this decision given that variances vary considerably between measures. This is not apparent in the current version of the manuscript. Again, I do not think that this is a serious problem, but in the interest of transparency I suggest adding this information and discussing the implications of performing SEM based on correlation matrices instead of covariance matrices

3. When playing around with the data a bit, I noted that the main problem for the correlated EF model causing its bad fit was the cross loading of the Trail-Making Task on the updating factor. This is the same cross-loading later implemented in the model including just shifting and updating. Why did the authors choose to not include this cross loading in the model with all three executive functions given that this would have achieved a good fit? This would then offer the possibility to include inhibition in their analysis as a predictor of individual differences in the CIE. I know that there were additional reasons to drop inhibition, but maybe including the cross-loading can mitigate problems with including inhibition in the SEM analysis.

4. Related to my previous point, the authors could have dropped the flanker task as an indicator of inhibition and constrained the loading of Stroop & go-no-go to be equal for the inhibition factors. That way the inhibition factor would be still locally identified, and the authors could present results with respect to relationship of inhibition with individual differences in the CIE.

5. I was surprised to see that the reliability of the differences scores for inhibition was so good, especially for the Stroop and flanker task. Could the authors share the full data for estimating these reliabilities and provide more information how reliability was estimated. I am familiar with omega in the context of CFA but computing it for item indicators is less common. I assume that the authors followed the method suggested by Hayes & Coutts (2020), but a more explicit description of the procedure would aid the reader in following the reliability estimation. Given the number of trials per condition (48 or 60) and results from previous research (for a summary see von Bastian et al., 2020) I would like to see some more information regarding these estimates to really trust them.

6. I am not entirely convinced that the additional exploratory factor analysis adds to the robustness of the results. I agree that SEM estimates are unstable with smaller sample sizes, especially if factor reliabilities are low. But the same problem applies to estimates for factor analysis that rely on correlation/covariance estimates. These are also not stable in smaller sample sizes and when correlations are low (Schönbrodt & Perugini, 2013). Therefore, the same critique could be used for the results of the exploratory factor analysis. Maybe I am missing a point here, in this case please elaborate why you think that EFA results should be more robust than the SEM results.

References

Frischkorn, G. T., von Bastian, C. C., Souza, A. S., & Oberauer, K. (2022). Individual differences in updating are not related to reasoning ability and working memory capacity. Journal of Experimental Psychology: General. https://doi.org/10.1037/xge0001141

Hayes, A. F., & Coutts, J. J. (2020). Use Omega Rather than Cronbach’s Alpha for Estimating Reliability. But…. Communication Methods and Measures, 14(1), 1–24. https://doi.org/10.1080/19312458.2020.1718629

Schönbrodt, F. D., & Perugini, M. (2013). At what sample size do correlations stabilize? Journal of Research in Personality, 47(5), 609–612. https://doi.org/10/f496x4

von Bastian, C. C., Blais, C., Brewer, G. A., Gyurkovics, M., Hedge, C., Kałamała, P., Meier, M. E., Oberauer, K., Rey-Mermet, A., Rouder, J. N., Souza, A. S., Bartsch, L. M., Conway, A. R. A., Draheim, C., Engle, R. W., Friedman, N. P., Frischkorn, G. T., Gustavson, D. E., Koch, I., … Wiemers, E. A. (2020). Advancing the understanding of individual differences in attentional control: Theoretical, methodological, and analytical considerations. PsyArXiv, 1–81.

Wilhelm, O., Hildebrandt, A., & Oberauer, K. (2013). What is working memory capacity, and how can we measure it? Frontiers in Psychology, 4, 433. https://doi.org/10/gd3vs7

Reviewer #2: This is a very interesting paper and I enjoyed reading it. It’s great to see a systematic examination of the relationship between EF/WM and the CIE that draws on contemporary models of WM such as Miyake’s account. The paper is nicely structured and well written, and the study design and analysis are appropriate for addressing the hypothesis/predictions. I don’t think there is a great deal that needs be to done to revise the paper – the comments that follow raise a few issues that I think can be dealt with relatively little effort. I’m cognisant of the fact that there’s not a lot of space available to add more content.

1. The relationship between the overview of the two models and the hypothesis and predictions felt a little loose. The way I read the introduction felt like it was setting up the study to play off the updating and inhibition accounts. Indeed the Discussion suggests that the findings favours the former. Yet the hypothesis and its associated predictions don’t make any mention of a test of the claims of the models. It would be nice to see a stronger link forged.

2. I found it a bit odd that the N-back tasks were described as measures of updating without additional justification. Admittedly there is some face validity for thinking this might be the case but I think it needs to be argued for – especially in light of the debate over what the N-back actually measures and whether it is a WM task at all.

Redick, T. S., & Lindsey, D. R. B. (2013) https://doi.org/10.3758/s13423-013-0453-9

Frost, A., Moussaoui, S., Kaur, J., Aziz, S., Fukuda, K., & Niemeier, M. (2021). https://doi.org/10.1016/j.actpsy.2021.103398

Kane, M. J., Conway, A. R. A., Miura, T. K., & Colflesh, G. J. H. (2007). https://doi.org/10.1037/0278-7393.33.3.615

It would be useful to provide some justification for using the N-back tasks this way. This is especially important given the fact the updating latent variables in the various analyses are strongly (mostly?) ‘driven’ by the N-back measures. If the validity of the N-backs as updating measures is in question then it does raise the questions as to what exactly it is that is explaining variance in the CIE.

3. Although it’s clear that the CIE has been used in individual differences (ID) research before I think there is a question as to whether it is an unproblematic ID measure. The paradigm that is used to generate the CIE measure used in the study is essentially an experimental one (à la (Johnson & Seifert, 1994) where the focus is on creating a task that is best distinguishes different experimental conditions/groups (accentuates between group variation) rather than one that preserves inter-individual variation. Thus, I wonder whether it might suffer from the same kinds of problems other experimental measures (e.g., ‘classic’ task switching and even the Stroop – see Rouder & Haaf, 2019) face when imported into ID research (typically an attenuation of covariance between the ID measures – often to the point where correlations etc are not statistically significant. This in turn can have all sorts of consequences for the likes of CFA, SEM and so on). If this is the case it is unclear to me how it might be influencing the relationships explored in the current study. Might a better ID CIE measure result in a different pattern of results on the SEM? Obviously these questions are beyond the scope of a revision of the paper – but I’d suggest that a brief discussion of the suitability of CIE measure for ID research would be helpful and assure the reader that limitations of the measure don’t mean we might be missing something important.

Hedge, C., Powell, G., & Sumner, P. (2018). https://doi.org/10.3758/s13428-017-0935-1

Rouder, J., Kumar, A., & Haaf, J. M. (2019). https://doi.org/10.31234/osf.io/3cjr5

Rouder, J. N., & Haaf, J. M. (2019). https://doi.org/10.3758/s13423-018-1558-y

4. The same issue could arise for the flanker task (a classic experimental task) and even the Stroop task (although the Stroop is widely used as an ID measure it is does seem to a strongly accentuate between groups variation and attenuate inter-individual variation). I do wonder whether the problem with the inhibition latent variable could possibly be related to the suitability of some of these tasks as ID measures.

5. Finally, a pretty minor but ‘big picture’ comment: it seems to me that it’s a bit of a stretch to argue that longer term, real world CIE effects, like the 5G example given in the introduction, are a ‘simple’ matter of poor WM/EF abilities (whether updating or inhibition issues). Unlike lab-based CIE tasks, in these real world situations it can be some time before disconfirming information comes to light. There seems to be an implication that the ‘action’ occurs at the point at which disconfirming information becomes available. But what if this doesn’t happen for weeks or months? Is the mental model ‘frozen’ until new information arrives? I’d expect not. I don’t think we can simply assume that people’s mental models are stable over longer periods of time – indeed, I think it’s quite possible (via confirmation bias type processes) that people seek out ‘confirming’ information and/or develop model-preserving strategies that can make updating or inhibition decreasingly likely to be effective in changing a person’s mental model (or selecting the better one). So the question is, how big a sway do WM/EF abilities have over belief change in these longer term, real world situations compared to lab-type CIE scenarios? Are they still important or do they become decreasingly important? I suspect WM/EF abilities also play a role in moderating the role of confirmation bias processes and the degree to which people adopt a ‘monological belief system’ about conspiracy theories and other potentially problematic beliefs. Importantly, I don’t think this is a huge problem for the paper – it’s fine to examine a ‘short-term CIE’. What we’ve learnt from the current paper is valuable – I just don’t think it necessarily can explain the kind of example used at the beginning of the paper.

6. PLOS authors have the option to publish the peer review history of their article (what does this mean?). If published, this will include your full peer review and any attached files.

Reviewer #1: No

Reviewer #2: **Yes: **Stephen R. Hill

---

## [Author Response · Author response to Decision Letter 0]

3 Feb 2023

Please find our responses to reviewers in the "Response to Reviewers" document provided with our resubmission.

---

## [Decision Letter · Decision Letter 1]

21 Mar 2023

Executive function and the continued influence of misinformation: A latent-variable analysis

PONE-D-22-17797R1

Dear Dr. McIlhiney,

We’re pleased to inform you that your manuscript has been judged scientifically suitable for publication and will be formally accepted for publication once it meets all outstanding technical requirements. There was one minor comment from Reviewer 1, but I believe you can adjust this during the proofreading process. I would like to thank the reviewers for their exceptional work evaluating this manuscript. I hope it makes a fine contribution to the literature on the misinformation effect, which is becoming a topic more and more relevant in our current context.

Kind regards,

Alessandra S. Souza, Ph.D.

Academic Editor

PLOS ONE

Additional Editor Comments (optional):

Reviewers' comments:

Reviewer's Responses to Questions

**Comments to the Author**

1. If the authors have adequately addressed your comments raised in a previous round of review and you feel that this manuscript is now acceptable for publication, you may indicate that here to bypass the “Comments to the Author” section, enter your conflict of interest statement in the “Confidential to Editor” section, and submit your "Accept" recommendation.

Reviewer #1: All comments have been addressed

Reviewer #2: All comments have been addressed

2. Is the manuscript technically sound, and do the data support the conclusions?

Reviewer #1: Yes

Reviewer #2: Yes

3. Has the statistical analysis been performed appropriately and rigorously? 

Reviewer #1: Yes

Reviewer #2: I Don't Know

4. Have the authors made all data underlying the findings in their manuscript fully available?

Reviewer #1: Yes

Reviewer #2: Yes

5. Is the manuscript presented in an intelligible fashion and written in standard English?

Reviewer #1: Yes

Reviewer #2: Yes

6. Review Comments to the Author

Reviewer #1: Overall, the authors addressed my concerns raised in my first review. The clarifications on the SEM analyses also resolved my question regarding the use of standardized observed variables. Which I now see the authors did not do and my reproduction of their results with using standardization for scaling and unstandardized observed indicators converges with the reported results. There is one minor points left that should be addressed prior to publication, but generally I think this manuscript is ready for publication.

1. On p. 17 the authors state that reliability was good except for the Keep-Track and CIE tasks. However, Table 1 indicates that reliability was low also for the Stroop and the Category Switch task.

Reviewer #2: I'm happy with the authors' responses to my comments. There seems to be consensus amongst the review reports that WM measurement and conceptualisation (e.g., being able to distinguish updating and storage contributions to scores; measurement of IDs) are complex and complicated and consequently some caution is required when drawing conclusions. The limitation statement about the N-back and the reference to the McIlhiney et al. (2022) paper on the CIE paradigm's ID measuring qualities are sufficient to keep me happy (Incidentally - I really appreciate being alerted to some useful references on these subjects!).

I'd like to note that, unlike Reviewer 1, I did not attempt to reproduce the statistical analyses but I'm glad to see that there were only a couple of minor requests for clarification/additional reporting and that there was no problem with reproducing the findings.

7. PLOS authors have the option to publish the peer review history of their article (what does this mean?). If published, this will include your full peer review and any attached files.

Reviewer #1: No

Reviewer #2: **Yes: **Stephen Robert Hill

---

## [Editor Report · Acceptance letter]

28 Mar 2023

PONE-D-22-17797R1 

Executive Function and the Continued Influence of Misinformation: A Latent-Variable Analysis 

Dear Dr. McIlhiney:

I'm pleased to inform you that your manuscript has been deemed suitable for publication in PLOS ONE. Congratulations! Your manuscript is now with our production department. 

Kind regards, 

on behalf of

Dr. Alessandra S. Souza 

Academic Editor

PLOS ONE